# Rapid Quantification of SARS-CoV-2-Neutralizing Antibodies Using Propagation-Defective Vesicular Stomatitis Virus Pseudotypes

**DOI:** 10.3390/vaccines8030386

**Published:** 2020-07-15

**Authors:** Ferdinand Zettl, Toni Luise Meister, Tanja Vollmer, Bastian Fischer, Jörg Steinmann, Adalbert Krawczyk, Philip V’kovski, Daniel Todt, Eike Steinmann, Stephanie Pfaender, Gert Zimmer

**Affiliations:** 1Institut für Virologie und Immunologie (IVI), Abteilung Virologie, CH-3147 Mittelhäusern, Switzerland; ferdinand.zettl@yahoo.de (F.Z.); philip.vkovski@vetsuisse.unibe.ch (P.V.); 2Department for Molecular and Medical Virology, Ruhr-Universität Bochum, Universitätsstrasse 150, D-44801 Bochum, Germany; toni.meister@rub.de (T.L.M.); daniel.todt@rub.de (D.T.); eike.steinmann@rub.de (E.S.); 3Institut für Laboratoriums- und Transfusionsmedizin, Herz- und Diabeteszentrum Nordrhein- Westfalen, Universitätsklinik der Ruhr-Universität Bochum, D-32545 Bad Oeynhausen, Germany; tvollmer@hdz-nrw.de (T.V.); bfischer@hdz-nrw.de (B.F.); 4Institute of Medical Microbiology, University Hospital of Essen, CH-45147 Essen, Germany; joerg.steinmann@uk-essen.de; 5Institute of Clinical Hygiene, Medical Microbiology and Infectiology, General Hospital Nürnberg, Paracelsus Medical University, D-90419 Nürnberg, Germany; 6Department of Infectious Diseases, West German Centre of Infectious Diseases, University Hospital Essen, University of Duisburg-Essen, D-45147 Essen, Germany; adalbert.krawczyk@uni-due.de; 7Institute for Virology, University Hospital of Essen, University of Duisburg-Essen, D-45147 Essen, Germany; 8Department of Infectious Diseases and Pathobiology (DIP), Vetsuisse Faculty, University of Bern, CH-3012 Bern, Switzerland

**Keywords:** SARS-CoV-2, neutralizing antibodies, pseudovirus, vaccine, passive immunization

## Abstract

Severe acute respiratory syndrome coronavirus type 2 (SARS-CoV-2, a new member of the genus *Betacoronavirus*, is a pandemic virus, which has caused numerous fatalities, particularly in the elderly and persons with underlying morbidities. At present, there are no approved vaccines nor antiviral therapies available. The detection and quantification of SARS-CoV-2-neutralizing antibodies plays a crucial role in the assessment of the immune status of convalescent COVID-19 patients, evaluation of recombinant therapeutic antibodies, and the evaluation of novel vaccines. To detect SARS-CoV-2-neutralizing antibodies, classically, a virus-neutralization test has to be performed at biosafety level 3, considerably limiting the general use of this test. In the present work, a biosafety level 1 pseudotype virus assay based on a propagation-incompetent vesicular stomatitis virus (VSV) has been used to determine the neutralizing antibody titers in convalescent COVID-19 patients. The neutralization titers in serum of two independently analyzed patient cohorts were available within 18 h and correlated well with those obtained with a classical SARS-CoV-2 neutralization test (Pearson correlation coefficients of r = 0.929 and r = 0.939, respectively). Most convalescent COVID-19 patients had only low titers of neutralizing antibodies (ND50 < 320). The sera of convalescent COVID-19 patients also neutralized pseudotype virus displaying the SARS-CoV-1 spike protein on their surface, which is homologous to the SARS-CoV-2 spike protein. In summary, we report a robust virus-neutralization assay, which can be used at low biosafety level 1 to rapidly quantify SARS-CoV-2-neutralizing antibodies in convalescent COVID-19 patients and vaccinated individuals.

## 1. Introduction

Severe acute respiratory syndrome coronavirus type 2 (SARS-CoV-2, a new member of the genus *Betacoronaviruses*, emerged in December 2019 in the city of Wuhan in China. SARS-CoV-2 is most closely related to bat SARS-like coronaviruses, suggesting that this new virus originally was a zoonosis [1,2,3]. In humans, SARS-CoV-2 is readily transmitted via aerosols or droplets. It infects cells expressing the human angiotensin-converting enzyme 2 (ACE2), which serves as a cellular receptor and is essential for viral entry [4,5]. Individuals of any age may be infected with SARS-CoV-2, but the elderly and persons with underlying morbidities particularly suffer from the disease (COVID-19), which is characterized by fever, coughing, and respiratory distress [2]. The airborne transmission, as well as the fact that humans do not possess pre-existing immunity to this virus, allowed SARS-CoV-2 to spread rapidly in the human population, causing a still ongoing pandemic.

Approved vaccines or antiviral drugs for the treatment of COVID-19 patients are not available yet. Due to the lack of other treatment options, passive immunization of severely ill patients with the plasma or serum of convalescent COVID-19 patients has been suggested [6]. The principle idea of this therapy is that virus-neutralizing antibodies in the serum/plasma of convalescent persons would help the patient to limit virus replication and to recover from the disease. Indeed, passive immunization is currently an approved treatment for patients suffering from infection with various other viral pathogens, including hepatitis A virus, hepatitis B virus, varicella zoster virus, cytomegalovirus, and rabies virus (post-exposure prophylaxis) [7,8,9]. In contrast to a vaccine, passive immunization is rapidly available, thereby representing a promising option to immediately treat severely ill COVID-19 patients.

Apart from passive immunization strategies, several vaccine candidates have been developed and are currently being evaluated in pre-clinical, as well as in advanced clinical, phases. It is generally believed that the induction of virus-neutralizing antibodies by vaccination will reduce the likelihood of (re)infection and development of severe disease, thus serving as a correlate of protection.

Enzyme-linked immunosorbent assays (ELISAs) for the detection of SARS-CoV-2 specific antibodies have become recently available; however, these assays do not provide any information about the levels of virus-neutralizing antibody titers. Therefore, a classical virus-neutralizing test with live SARS-CoV-2 has to be performed under biosafety level 3 conditions. The readout of this test is based on the virus-induced cytopathic effect, which takes 2 to 3 days to fully develop.

Vesicular stomatitis virus (VSV) is a negative-strand RNA virus which possesses a single type envelope glycoprotein (G) mediating viral entry. Previous studies have shown that G-deleted VSV can be complemented in trans with the envelope glycoproteins of even unrelated viruses, including coronaviruses [5,10,11,12]. Importantly, pseudotype viruses are susceptible to neutralization by antibodies directed to the foreign glycoprotein [13]. Since pseudotype viruses are restricted to a single round of infection, these pseudotype virus neutralization assays do not require enhanced biosafety measures. Moreover, the VSV genome may be easily manipulated in order to encode for reporter proteins, such as green fluorescent protein (GFP) and luciferase, which facilitates the detection of infected cells in a reasonable period of time. In this work, we pseudotyped VSV*ΔG(FLuc), a G-deleted VSV encoding both GFP and firefly luciferase [14], with the SARS-CoV-2 spike protein and used it for the detection of SARS-CoV-2 neutralizing antibodies in the sera of convalescent COVID-19 patients.

## 2. Materials and Methods

### 2.1. Cells and Virus

Vero E6 cells were kindly provided by Christian Drosten/Marcel Müller (Charite, Berlin, Germany) and maintained in Dulbecco’s minimal essential medium (DMEM; Life Technologies, Zug, Switzerland) supplemented with 10% fetal bovine serum (FBS) and non-essential amino acids (Life Technologies). Baby hamster kidney (BHK-21) cells were obtained from American Type Culture Collection (ATCC^®^ CCL-10; Manassas, VA, USA) and maintained in Glasgow’s minimal essential medium (GMEM, Life Technologies) supplemented with 10% FBS. BHK-G43, a transgenic BHK-21 cell clone expressing the VSV G protein in a regulated manner [15], was maintained in GMEM containing 5% FBS. I1-Hybridoma, which secrete a VSV-neutralizing antibody directed to the envelope glycoprotein G [16], were purchased from ATCC^®^ (CRL-2700^™^) and maintained in minimal essential medium (MEM, Life Technologies) supplemented with 15% FBS. Human embryo kidney (HEK) 293T cells (ATCC, CRL-3216^™^) were cultured using DMEM with 10% FBS. 

SARS-CoV-2 (SARS-CoV-2/München-1.1/2020/929) was kindly provided by Christian Drosten (Charité, Berlin, Germany) propagated on Vero E6 cells. SARS-CoV-2 (virus isolate UKEssen) was isolated from a patient sample in Essen, Germany.

### 2.2. Detection of SARS-CoV-2 by RT-qPCR

For the detection of SARS-CoV-2 virus, the Allplex™ SARS-CoV-2 specific RT-qPCR assay (Seegene (Seoul, Korea)) and the RealStar SARS-CoV-2 RT-PCR Kit (Altona Diagnostic Technologies (ADT), Hamburg, Germany) were used according to the manufacturers’ instructions. The samples were run on a CFX96™ real-time thermal cycler (Bio-Rad, Feldkirchen, Germany).

### 2.3. Collection of Serum/Plasma from Convalescent COVID-19 Patients

Patient cohort 1 samples were derived from 21 convalescent plasma donors. Sera and EDTA-plasma samples were either collected in 7.5 mL serum gel or K3 EDTA monovettes (Sarstedt, Nümbrecht, Germany) followed by centrifugation at 2500× *g* for 10 min. Citrate-plasma samples were collected by plasmapheresis from convalescent plasma donors as a side product of the collection process. In addition, serum samples were collected from 4 patients who were still under intensive medical care and included in this cohort (Table 1). Patient cohort 2 samples were obtained from 13 hospital staff members who had been tested positive for SARS-CoV-2 (Table 2). Serum samples were collected in 7.5 mL serum monovettes (Sarstedt, Nümbrecht, Germany) and subsequently processed by centrifugation at 2500× *g* for 10 min. 

All samples were collected in accordance with the German Act on Medical Devices (MPG guideline 98/79/EC) for the collection of human residual material to evaluate suitability of an in vitro diagnostic medical device (§24). The need for informed consent and ethical approval was waived since all materials used were residual from routine laboratory diagnostics.

### 2.4. ELISA

For detection of SARS-CoV-2 spike subunit 1-specific IgG or IgA, the Euroimmun enzyme-linked immunosorbent assays (ELISA) were used according to the manufacturer’s instructions (Euroimmun AG, Lübeck, Germany ^#^ EI 2606-9601 G and ^#^ EI 2606-9601 A). The Euroimmun ELISA has been recently validated with respect to specificity and sensitivity using COVID-19 patient sera [17]. Binding of antigen-specific antibodies was recorded at an optical density of 450 nm (OD_450_) and then divided by the OD_450_ of the calibrator provided with each ELISA kit to minimize inter-assay variation (ratio). Results of the immunoassay were classified into the three categories: negative (ratio < 0.8), borderline (ratio > 0.8 to ≤1.1), and positive (ratio > 1.1). 

### 2.5. Generation of Pseudotype Virus

VSV*∆G(FLuc), a G-deficient VSV encoding GFP and firefly luciferase, has been described previously [14] and was propagated on transgenic BHK-G43 cells expressing the VSV G protein after induction by mifepristone [15]. The trans-complemented particles (about 10^8^ focus-forming units per milliliter (ffu/mL) were stored at −70 °C in the presence of 5% FBS. Pseudotype viruses were titrated on BHK-21 cells in 96-well cell culture plates as described previously [14]. Virus titers ranged from 2500 to 10,000 ffu/mL.

For production of pseudotype virus, BHK-21 or alternatively HEK 293T cells were grown in 10-cm cell culture dishes and transfected with 10 µg of pCG1-SARS-CoV-2 (Wuhan-Hu-1) spike protein expression plasmid [5] or 10 µg of SARS-CoV-1 (Frankfurt-1) spike protein expression plasmid pCAGGS-SARS-S [5], both kindly provided by Stefan Pöhlmann (German Primate Center, Göttingen, Germany). Lipofectamine 2000 (Life Technologies, Zug, Switzerland) was employed as transfection reagent. At 20 h post-transfection, the cells were washed once with GMEM and subsequently inoculated for 60 min at 37 °C with VSV*∆G(FLuc) using an moi of 5 ffu/cell. The cells were washed twice with GMEM and subsequently incubated at 37 °C with GMEM containing 5% FBS and 10% conditioned cell culture medium of hybridoma cells secreting the VSV neutralizing monoclonal antibody Mab I1, which prevents carryover of VSV particles carrying the G protein rather than the CoV-S on their cell surface. At 20 h pi, the cell culture supernatant was harvested, cell debris removed by low-speed centrifugation (1000× *g*, 10 min, 4 °C), and the pseudotype viruses stored in aliquots at −70 °C. Taking advantage of the GFP reporter for detection of infected cells, the pseudotype viruses were titrated on Vero E6 cells, which are highly permissive to SARS-CoV-1 and SARS-CoV-2 [18].

### 2.6. Pseudotype Virus Neutralization Tests

Convalescent plasma/sera were incubated for 30 min at 56 °C in order to inactivate complement factors. For the pseudotype virus neutralization (PVN) test, twofold serial dilutions of the immune sera were prepared in DMEM cell culture medium. The diluted sera were added to 96-well cell culture plates (50 μL/well, quadruplicates for each dilution) and incubated for 60 min at 37 °C with pseudotype virus (50 μL/well containing 200 ffu). Vero E6 cell suspension (100,000 cells/mL was added to the wells (100 µL/well) and incubated for 16 h at 37 °C. The cell culture medium was aspirated and the cells lysed with 50 µL of luciferase lysis buffer (Promega, Dübendorf, Switzerland). The cell lysate (25 µL) was transferred to white microtiter plates before 25 µL of firefly luciferase ONE-Glo^TM^ substrate (Promega) was added and luminescence recorded using a GloMax^®^ plate reader (Promega). The reciprocal antibody dilution causing 50% reduction of the luminescence signal was calculated and expressed as pseudotype virus neutralization dose 50% (PVND_50_). 

### 2.7. SARS-CoV-2 Neutralization Test

Patient cohort 1 was analyzed using SARS-CoV-2 (SARS-CoV-2/München-1.1/2020/929), whereas, for patient cohort 2, SARS-CoV-2 (UKEssen) was utilized. Twofold serial dilutions of heat-inactivated immune sera/plasma were prepared in quadruplicates in 96-well cell culture plates using DMEM cell culture medium (50 µL/well). To each well, 50 µL of DMEM containing 200 tissue culture infectious dose 50% (TCID_50_) of SARS-CoV-2 were added and incubated for 60 min at 37 °C. Subsequently, 100 µL of Vero E6 cell suspension (100,000 cells/mL in DMEM with 10% FBS) were added to each well and incubated for 56 h at 37 °C. The cells were fixed for 1 h at room temperature with 4% buffered formalin solution containing 1% crystal violet (Merck, Darmstadt, Germany). Finally, the microtiter plates were rinsed with deionized water and immune serum-mediated protection from cytopathic effect was visually assessed. Neutralization doses 50% (ND_50_) values were calculated according to the Spearman and Kärber method [19]. 

### 2.8. Statistical Analysis 

The Pearson correlation coefficient r and linear regression was calculated using GraphPad Software (San Diego, CA, USA). Kruskal–Wallis test and Dunns’ multiple comparison test were performed to statistically analyze the impact of anti-coagulants on neutralizing antibody titers. 

### 2.9. Ethical Permission

All samples used in the study were analyzed after conventional diagnostic tests had been performed. The study did not include patient´s details and did not result in additional constraints for the patients. All analyses were carried out in accordance with approved guidelines.

## 3. Results

In order to determine virus-neutralizing antibody levels in sera collected from convalescent COVID-19 patients, we employed a biosafety level 1 compliant pseudotype virus neutralization (PVN) test using VSV*ΔG(FLuc), a propagation-incompetent vesicular stomatitis virus (VSV) that lacked the glycoprotein (G) gene but harbored the reporter genes green fluorescent protein (GFP) and firefly luciferase (FLuc) instead [14]. VSV*ΔG(FLuc) was propagated on VSV G protein expressing cells, resulting in G protein trans-complemented particles, which were then used to infect cells which transiently expressed the spike protein of either SARS-CoV-1 or SARS-CoV-2 (Figure 1). The infected cells, which were maintained in the presence of VSV G neutralizing antibody, produced infectious pseudotype particles that harbored the corresponding spike protein in the viral envelope. Infection of Vero E6 cells by pseudotype virus was inhibited by spike-specific neutralizing antibodies and was quantified taking advantage of the FLuc reporter protein. The serum dilution resulting in 50% reduction of luciferase reporter activity was calculated and expressed as pseudotype virus neutralization dose 50% (PVND_50_).

In the sera of convalescent COVID-19 patients, neutralizing antibodies were detected (Table 1, Figure 2a), which was not the case for a COVID-19 negative patient (Figure 2b). No significant differences were observed when comparing patient sera with EDTA-plasma (*p* > 0.9999) or citrate-plasma (*p* = 0.5530) or comparing EDTA-plasma with citrate-plasma (*p* > 0.9999), indicating that the collection method did not interfere with neutralizing antibody titer estimation (Figure 2c). Testing of 25 patients of various age (26 to 76 years old), who had recovered from COVID-19 with mild to moderate symptoms, revealed the presence of neutralizing antibodies with varying titers (Table 1). For comparison, serum samples were collected from 4 patients who were still under intensive medical care. SARS-CoV-2 spike-specific IgG and IgA antibodies were detected by ELISA in all patient sera, but levels seemed to vary considerably as indicated by IgG and IgA ratios (Table 1).

The PVND_50_ values varied considerably among the patients ranging from <10 to 1280 (Table 1). Interestingly, several patients (patients ^#^8, ^#^9, ^#^14, ^#^16, ^#^22–25) who had high neutralizing antibody titers also showed high ELISA IgA ratios, although patients ^#^1 and ^#^15 did not. Surprisingly, patients who were still under intensive medical care belonged to this group of high responders.

Most sera also showed neutralizing activity against pseudotype viruses bearing the SARS-CoV-1 spike protein (Table 1). Although SARS-CoV-1 neutralizing antibody titers were mostly lower than those directed to SARS-CoV-2, there was a significant correlation between SARS-CoV-1 and SARS-CoV-2 spike specific neutralization titers (Pearson r = 0.7811, *p* < 0.0001). Since it is unlikely that these patients had been in contact with SARS-CoV-1 in the past, this inhibitory activity is likely due to cross-reaction of SARS-CoV-2 specific antibodies with the spike protein of SARS-CoV-1, both spike proteins sharing about 77% homology based on their primary amino acid sequence.

In order to evaluate the specificity of the PVN test we also investigated whether immune sera from convalescent COVID-19 patients would neutralize VSV*ΔG(FLuc) trans-complemented with the VSV G protein. Since VSV is endemic in the Americas, European citizen probably have rarely been in contact with this arthropod-borne virus. As expected, most COVID-19 patients had no neutralizing antibodies directed to the VSV G antigen (Table 1). However, patients ^#^2 and ^#^3 showed PVND_50_ values of 160, indicating that these two persons had been infected once with VSV serotype Indiana. There was no significant correlation between neutralizing antibody titers to VSV G protein and SARS-CoV-2 spike protein (Pearson r = −0.0138, *p* = 0.9478).

All sera of the COVID-19 patient cohorts were also tested at biosafety level 3 with a standard SARS-CoV-2 neutralization test. With this test, the reciprocal antibody dilution that fully protected 50% of Vero E6 cells from virus-induced cytopathic effect (CPE) at 56 h post-infection (pi) was determined (neutralization dose 50%, ND_50_) (Table 1). Although PVND_50_ and ND_50_ values were based on a completely different readout (luciferase reporter activity vs. CPE), the neutralizing antibody titers as determined by these two methods correlated very well, as indicated by Pearson r values of 0.929 (Figure 3). In order to evaluate the robustness of the PVN test, a distinct cohort of hospital staff members who have been tested positive for SARS-CoV-2 by RT-PCR or ELISA was also investigated (Table 2). The correlation between the PVND_50_ and ND_50_ again correlated very well (r = 0.939), also the PVND_50_ tends to overestimate the titer in this cohort, as the slope is with 0.686 slightly flatter compared to the patient cohort (0.955) (Figure 3).

## 4. Discussion

There is evidence from experimental vaccination that neutralizing antibodies protect animals from SARS-CoV-1 infection and disease [20]. In the absence of any other therapeutic options, passive immunization using plasma of convalescent COVID-19 patients might represent a promising and immediately available therapy for the treatment of severely ill COVID-19 patients [21,22]. Similarly, virus-neutralizing antibodies are probably the most important mediators of protection in vaccinated individuals.

In the present work, we used a VSV pseudotype system to detect and to titrate SARS-CoV-2-neutralizing serum antibodies from convalescent COVID-19 patients. In contrast to SARS-CoV-2, which has to be handled at biosafety level 3 (BSL-3), the VSV*ΔG(FLuc) vector, which has been used for pseudotyping, can be handled at BSL-1 conditions, thus being the case in most diagnostic laboratories. Since the glycoprotein G gene has been deleted from the VSV*ΔG(FLuc) genome, the virus is unable to produce any progeny and thus is restricted to a single cycle of infection. However, due to a strong expression of the firefly luciferase (FLuc) reporter gene encoded by the viral genome, the PVN test could be already read 18 h post-infection, which is significantly faster than the 56 h that were required to read the classical neutralization test with live SARS-CoV-2. We used the FLuc reporter as readout as it allowed us to easily quantify antibody-mediated inhibition of infection. The FLuc reporter may also allow high throughput screening of high numbers of sera for the presence of SARS-CoV-2-neutralizing antibodies if titration is not necessary. Despite faster performance compared to the classical virus neutralization test, the PVN test did not lack behind the classical virus neutralization test in terms of sensitivity and specificity.

For the therapy of severely ill COVID-19 patients, plasma is normally collected from convalescent donors ≥ 14 days after resolution of mild to moderate symptoms [22]. According to an EU guidance for COVID-19 convalescent plasma collection and transfusion, donor plasma should have SARS-CoV-2-neutralizing antibodies with ND_50_ values equal or higher than 320 [23]. In our cohorts of convalescent patients several patients showed ND_50_ values below this critical threshold value. It might be therefore necessary to screen many convalescent sera/plasma before the rare high responders are found who could serve as appropriate plasma donors. Another important criterion is the amount of IgA in the donor plasma as this isotype (as well as IgM) can be secreted into the mucosal tissues of the respiratory tract where SARS-CoV-2 replicates. Interestingly, the majority of patients showing high neutralizing antibody titers also had high IgA ELISA ratios. Since IgA dimers have been shown to neutralize influenza viruses efficiently due to their quaternary structure [24], we hypothesize that spike-specific IgA dimers may be more potent in neutralizing SARS-CoV-2 than IgG. The same might hold true for IgM pentamers. It is tempting to speculate that the increased valency of IgA and IgM would lead to improved virus neutralization by mediating enhanced aggregation of virions or bridging of spikes on individual virions. However, this interesting aspect needs to be verified in future studies. Unfortunately, both IgA and IgM are not persisting for prolonged time and therefore may not contribute to long-lasting immunity.

In addition to the testing of potential plasma donors for SARS-CoV-2-neutralizing antibodies, COVID-19 patients should be tested, as well, before they receive the donor plasma. Our study suggests that at least some COVID-19 patients may already have high levels of neutralizing antibodies in the blood. In particular, if the patient is under prolonged intensive medical care treatment, the apparent symptoms might be immune driven and not due to direct virus-mediated damage to the host [25]. Thus, the timing of the therapy might be critically important for the success of the therapy.

For several viral diseases, such as influenza [26] and measles [27], virus-neutralizing antibodies are a well-established correlate of protection. In contrast, the role of serum antibodies in the protection of humans from SARS-CoV-2 infection is not well understood. Passive transfer of murine serum to naïve mice protected these animals from SARS-CoV-1 challenge infection, suggesting that neutralizing antibodies alone may sufficiently control virus replication [20,28]. However, the antibody levels that would mediate sufficient protection of humans still need to be defined. This aspect is also relevant given that many SARS-CoV-2 vaccines, which are currently developed, are valued according to the levels of neutralizing antibodies they induce.

Several convalescent COVID-19 patients showed only low levels of neutralizing serum antibodies. At present, it is not known whether these neutralizing antibody levels are sufficiently high in order to protect the convalescent patients from repeated infection with SARS-CoV-2. Moreover, it is not known how these antibody levels would persist with time or how quickly they will drop. To address all these important questions, the pseudotype virus neutralization test described in this work will be very helpful.

## 5. Conclusions

In the present work, a robust pseudotype virus neutralization test has been developed which allows the rapid and sensitive quantification of SARS-CoV-2-neutralizing antibodies at biosafety level 1. Since the neutralizing antibody titers produced by the pseudotype virus assay correlated very well with those produced by a standard SARS-CoV-2 neutralization test which was run at biosafety level 3, this pseudotype virus assay will be useful to screen human B cell clone supernatants for neutralizing antibodies, evaluation of recombinant nanobodies, testing of immune sera from convalescent COVID-19 patients, and evaluation of experimental vaccines. All this will contribute to the control of the current SARS-CoV-2 pandemic.

## Figures and Tables

**Figure 1 vaccines-08-00386-f001:**
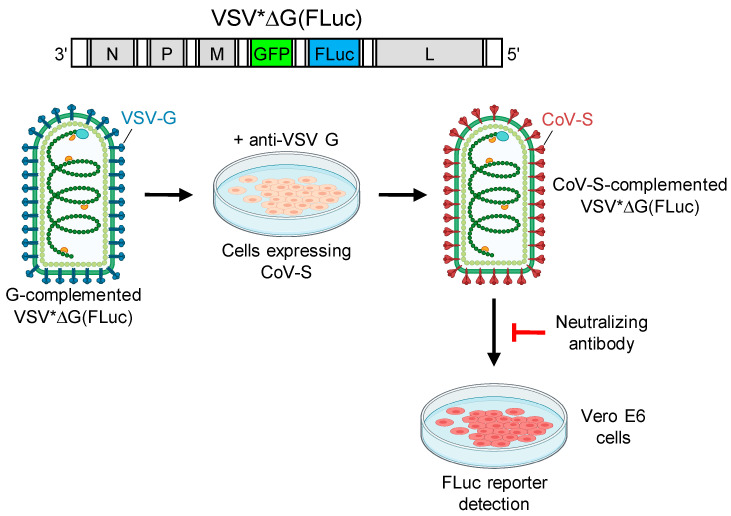
Pseudotyping of the vesicular stomatitis virus (VSV)*ΔG(FLuc) reporter virus with the SARS-CoV-2 spike protein. The RNA genome of this virus encodes for the nucleoprotein (N), the phosphoprotein (P), the matrix protein (M), the green fluorescent protein (GFP), the firefly luciferase (FLuc), and the large RNA polymerase (L), but lacks the viral glycoprotein (G) gene. Propagation of the G-deficient VSV*ΔG(FLuc) on cells expressing the VSV G protein results in trans-complemented viruses (G-complemented VSV*ΔG(FLuc), depicted in blue), which are used to infect cells transiently expressing either the SARS-CoV-1 or the SARS-CoV-2 spike protein. In the presence of the VSV-neutralizing antibody Mab I1 (anti-VSV G), pseudotype VSV*ΔG(FLuc) particles bearing the corresponding CoV spike protein in the envelope (depicted in purple) are released. Infection of Vero E6 cells by these pseudotype particles is inhibited by spike-specific neutralizing antibodies. The pseudotype virus infection rate is determined by measuring the FLuc reporter activity in the cell lysates 18 h post-infection. Image created with BioRender.com.

**Figure 2 vaccines-08-00386-f002:**
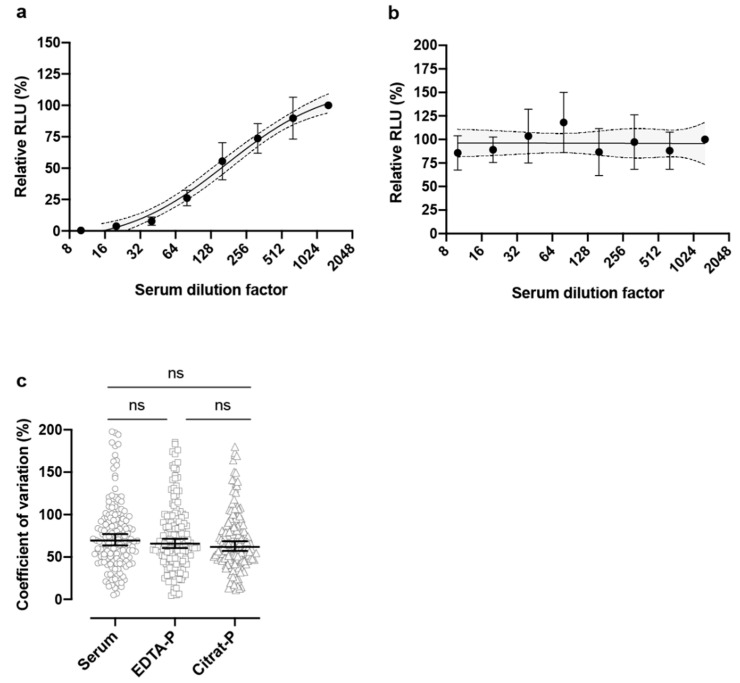
Quantification of SARS-CoV-2 spike-specific neutralizing antibody titers by the PVN test. SARS-CoV-2 spike-pseudotyped VSV*ΔG(FLuc) was incubated for 1 h with serial dilutions of serum from a convalescent COVID-19 patient (**a**) or from a COVID-19-negative cancer patient (**b**) prior to infection of Vero E6 cells. FLuc reporter activity in cell lysates was determined 18 h post-infection. The relative light units (RLU) detected in cells following pseudotype virus infection in the absence of patient serum were set to 100%. Mean values, standard deviation, and confidence intervals of quadruplicate analysis are shown. (**c**) Impact of anti-coagulants on neutralizing antibody titers. Serum, EDTA plasma, or citrate plasma of 20 patients were analyzed by the pseudotype virus neutralization (PVN) test for neutralizing antibodies directed to the SARS-CoV-2 spike protein and the coefficient of variation of four technical replicates per dilution was calculated. ns: not significant.

**Figure 3 vaccines-08-00386-f003:**
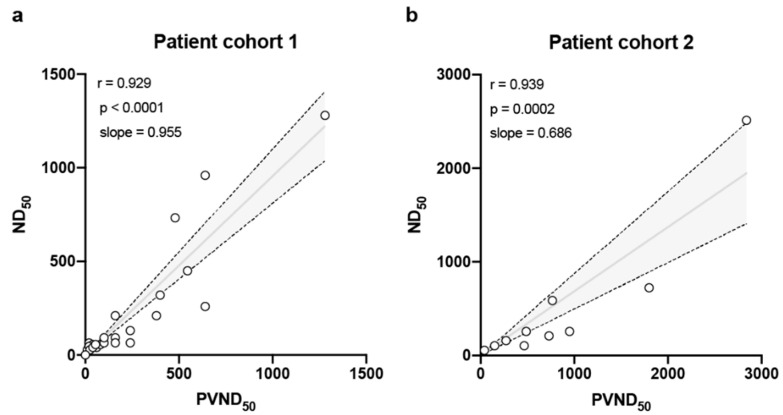
Comparison of the PVN assay with the classical virus neutralization test using live SARS-CoV-2. The pseudotype virus neutralization dose 50% (PVND_50_) and the SARS-CoV-2 neutralizing dose 50% (ND_50_) were determined in the sera from two independent cohorts of individuals that had been tested positive for SARS-CoV-2 infection by RT-PCR and/or ELISA. (**a**) Convalescent plasma donors and (**b**) cohort of hospital staff members. Line and band indicate slope of linear regression and respective confidence intervals; r = Pearson correlation coefficient.

**Table 1 vaccines-08-00386-t001:** Quantification of severe acute respiratory syndrome coronavirus type 2 (SARS-CoV-2)-neutralizing antibodies in sera from convalescent plasma donors (cohort 1).

Patient No.	Age (y)	Patient Status	Disease Severity	Virus Detection (RT-PCR)	IgG Ratio	IgA Ratio	ND_50_ ^a^	PVND_50_ ^b^		
								SARS-CoV-2	SARS-CoV-1	VSV G
1	45	Convalescent	Mild-Moderate	+ ^c^	4.2	7.9	64	20	<10	<10
2	64			+	4.7	5.1	92	160	20	160
3	64			+	3.9	5.0	65	160	30	160
4	53			+	1.4	0.7	28	10	<10	<10
5	53			+	1.3	1.3	40	60	<10	<10
6	42			+	2.2	2.0	56	40	40	<10
7	49			+	2.2	4.4	65	160	80	<10
8	49			+	3.9	>8	210	380	50	15
9	49			+	1.7	>9	259	640	40	40
10	34			+	2.7	0.8	46	20	<10	<10
11	48			+	2.0	3.3	46	20	<10	10
12	48			+	1.6	4.7	28	25	<10	<10
13	34			+	7.2	3.3	130	80	<10	<10
14	38			+	1.0	6.8	320	400	40	<10
15	29			n.d. ^d^	1.3	7.4	56	80	80	<10
16	27			+	3.3	>8	210	160	160	<10
17	53			−	2.4	1.3	<10	<10	<10	<10
18	46			+	2.3	2.7	65	100	<10	<10
19	47			+	1.3	0.1	40	40	<10	<10
20	26			+	3.3	4.2	56	55	10	<10
21	26			+	2.7	5.6	92	100	80	<10
22	75	Intensive medical care	Severe	n.d.	1.9	>9	960	640	640	<10
23	66			+	11.6	>8	1280	1280	640	<10
24	56			n.d.	13	>8	733	480	640	<10
25	59			n.d.	8.8	>9	450	546	560	<10

^a^ Neutralization dose 50% (ND50); ^b^ Pseudotype virus neutralization dose 50% (PVND50); ^c^ “+”, detected, “-“, not detected; ^d^ n.d., not determined.

**Table 2 vaccines-08-00386-t002:** Quantification of SARS-CoV-2-neutralizing antibodies in sera collected from hospital workers (cohort 2).

Patient No.	Age (y)	Patient Status	Disease Severity	Virus Detection (RT-PCR)	IgG Ratio	IgA Ratio	SARS-CoV-2	
							ND_50_ ^a^	PVND_50_ ^b^
1	37	Convalescent	Mild	+ ^c^	1.3	n.d. ^d^	320	158
2	34		Mild	n.d.	1.2	1.8	<20	<20
3	46		Mild	+	6.8	6.3	480	257
4	32		Mild	−	6.3	4.7	560	588
5	40		Mild	+	6.6	>10.3	800	257
6	32		Mild	+	5.1	4.8	50	56
7	63		Mild	+	13.2	>10.3	2560	2512
8	62		Mild	+	4.7	1.5	900	209
9	42		Mild	−	5.0	4.6	120	105
10	48		Mild	+	2.5	<0.8	<20	<20
11	47		Mild	+	5.1	8.7	240	105
12	49		Mild	+	3.3	5.9	1920	724
13	63		Mild	−	2.5	<0.8	<20	<20

^a^ Neutralization dose 50% (ND50); ^b^ Pseudotype virus neutralization dose 50% (PVND50); ^c^ “+”, detected, “-“, not detected; ^d^ n.d., not determined.

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
