# Peer review of "Rapid Quantification of SARS-CoV-2-Neutralizing Antibodies Using Propagation-Defective Vesicular Stomatitis Virus Pseudotypes"

_vaccines, 2020, doi:10.3390/vaccines8030386_

Round 1

Reviewer 1 Report

The manuscript “Rapid quantification of SARS-CoV-2-neutralizing antibodies using propagation-defective vesicular stomatitis virus pseudotypes” by Zettl et al. reported a new and fast detection and quantification method of SARS-CoV-2-neutralizing antibodies, by using psudotyped virus, which could be performed in a biosafety level 1, instead of traditional virus-neutralization test that is need to be performed in a biosafety level 3. The method is based on measuring reporter gene activity, because the propagation-incompetent vesicular stomatitis reporter virus deficient in glycoprotein (G), but instead encodes GFP and luciferase, and the infected cells produced infectious pseudotype particles that harbored SARS-CoV spike protein in the viral envelope. This method has previously been used for detecting type I interferon, main merits are fast, safe, and accurate.

This is a well-written manuscript. The manuscript is logic and quite thorough (included proper controls). This study is timely, should be beneficial for many labs only have biosafety level 1. Addressing the following issues should improve the manuscript:

The authors briefly described the methods at the beginning of the results part (line 187 to 199), in line 191, “VSV*DG(Fluc) was propagated on VSV G protein expressing 191 cells”, however, in Figure 1., the authors indicated VSV-G on the surface of viral genome, which is confusing, it should be VSVDG, therefore, should not be included. The authors might need to either remove VSV-G part (blue color) from the surface, or add more description after VSV-G (deleted etc). The authors want to illustrate VSV-G was complemented by CoV-S in Figure 1, but Figure 1 need to be modified based on above suggestion.

It would be nice to show If the authors have western blot result of CoV-S, which indicates CoV-S actually expressed in these cells. If not, that is fine too.

Figure 2c, please add p-vales.

Figure 3, it is convincing that there is a high correlation between ND50 and PVND50.

Reviewer 2 Report

            The manuscript by Zettl et al. describes an assay for a subset of antibodies to SARS-CoV-2 that is intended to represent the same set of antibodies measured in various assays of “neutralization” of infectious SARS-CoV-2.  The principal advantage of this assay is that it utilizes a replication-defective pseudotyped surrogate of SARS-CoV-2, and therefore can be performed without the delay, expense, and risk of live-virus assays.  Another advantage is a more rapid result, as the assay described here can performed in 1 day instead of 4 or more.  This assay, then, is offered as a correlate of a presumptive statistical correlate of immunity; that is, even infectious-virus neutralization assays are uncertain mechanistic correlates of immunity for SARS-CoV-2.  Still, the importance of in vitro neutralization assays represents mainstream understanding in virology, and improved (reliable, readily available) serological assays for functional immunity to SARS-CoV-2 are badly needed.  This manuscript is a worthwhile contribution to that overall effort.

            This manuscript does not place emphasis on mechanisms of neutralization, or how a range of assays—all called neutralization assays—may differ substantially in what they measure.  Here, the reader is left to assume (logically enough) that the assay measures only the antibody-mediated inhibition of binding and entry by the pseudotyped virus; this contrasts with the internal benchmark of the ND50 assay, in which antibodies may inhibit or slow the spread of virus (entry, exit, and other possibilities) over the course of several days.  The intriguing possibility is raised that the increased valency of IgA and IgM may lead to improved neutralization in vitro (and in vivo?), but it dangles without further explanation:  is the phenomenon (if real and reproducible) due to virion aggregation? to bridging of spikes on individual virions? to accessory molecules like complement?  What are the implications for lasting immunity and immunopathology?

            So, mostly, the manuscript describes an assay, and that too is useful.  It does not go so far as to provide validation of sensitivity and specificity of the assay, which can only come with additional patient samples and far more controls.

            The above narrative addresses a matter between authors and Journal, i.e. whether the manuscript meets this journal’s requirements for novelty, significance, and likely reader interest.  Beyond that, I suggest a few small improvements to enhance readability and make context more transparent:

            1) The VSV-neutralizing Mab I1 should be referenced or better described.

            2)  Briefly, describe the degree to which carryover of VSV G is otherwise problematic (necessitating its neutralization); how critical is this step, or this particular Mab, in assay reliability?  My brief perusal of cited references did not clarify this, and readers should not be obliged to hunt.

            3) It would be helpful to describe (briefly, parenthetically) the range of titers typically obtained in the two production steps: the VSV-G-containing virus stock and the pseudotyped (SARS spike-containing) preparation?   

            4) At least once, note that “expressing the VSV G protein in a regulated manner ” refers (presumably) to mifepristone. Readers should not have to hunt previous references and product literature to have a clear idea what’s happening.  Similarly, since product literature has no permanence in scientific literature, it would be helpful to describe a little more about the ELISA, including what the provider currently asserts for sensitivity and specificity.

            5) Around line 153, it would be helpful to note why Vero cells are used; presumably (like kidney cells in general, though these are African green) they have ACE-2 receptors required by SARS-CoV-2.  Reference?

            6) Regarding heat-inactivation of complement (line 156), do experimental data (e.g., not shown) show complement to be problematic in assay reproducibility, or is this an unexamined standard precaution? Does re-addition of complement improve sensitivity of neutralization assay?  (not necessary to do additional experiments, but say whether it’s known).

            7) I invite authors to embellish narratives in response to the first three paragraphs above, but do not require it. 
